# Impact of Drug-Gene-Interaction, Drug-Drug-Interaction, and Drug-Drug-Gene-Interaction on (es)Citalopram Therapy: The PharmLines Initiative

**DOI:** 10.3390/jpm10040256

**Published:** 2020-11-28

**Authors:** Muh. Akbar Bahar, Pauline Lanting, Jens H. J. Bos, Rolf H. Sijmons, Eelko Hak, Bob Wilffert

**Affiliations:** 1Department of PharmacoTherapy, -Epidemiology & -Economics, University of Groningen, 9713 AV Groningen, The Netherlands; h.j.bos@rug.nl (J.H.J.B.); e.hak@rug.nl (E.H.); b.wilffert@rug.nl (B.W.); 2Faculty of Pharmacy, Universitas Hasanuddin, Makassar 90245, Indonesia; 3Department of Genetics, Groningen, University Medical Center Groningen, University of Groningen, 9713 AV Groningen, The Netherlands; p.lanting@umcg.nl (P.L.); r.h.sijmons@umcg.nl (R.H.S.); 4Department of Clinical Pharmacy and Pharmacology, University Medical Center Groningen, University of Groningen, 9713 AV Groningen, The Netherlands

**Keywords:** (es)citalopram, drug-gene-interaction, drug-drug-interaction, drug-drug-gene-interaction, the PharmLines initiative

## Abstract

We explored the association between CYP2C19/3A4 mediated drug-gene-interaction (DGI), drug-drug-interaction (DDI) and drug-drug-gene-interaction (DDGI) and (es)citalopram dispensing course. A cohort study was conducted among adult Caucasians from the Lifelines cohort (167,729 participants) and linked dispensing data from the IADB.nl database as part of the PharmLines Initiative. Exposure groups were categorized into (es)citalopram starters with DGI, DDI and DDGI. The primary outcome was drug switching and/or dose adjustment, and the secondary was early discontinuation after the start of (es)citalopram. Logistic regression modeling was applied to estimate adjusted odd ratios with their confidence interval. We identified 316 (es)citalopram starters with complete CYP2C19/3A4 genetic information. The CYP2C19 IM/PM and CYP3A4 NM combination increased risks of switching and/or dose reduction (OR: 2.75, 95% CI: 1.03–7.29). The higher effect size was achieved by the CYP2C19 IM/PM and CYP3A4 IM combination (OR: 4.38, 95% CI: 1.22–15.69). CYP2C19/3A4 mediated DDIs and DDGIs showed trends towards increased risks of switching and/or dose reduction. In conclusion, a DGI involving predicted decreased CYP2C19 function increases the need for (es)citalopram switching and/or dose reduction which might be enhanced by co-presence of predicted decreased CYP3A4 function. For DDI and DDGI, no conclusions can be drawn from the results.

## 1. Introduction

Selective serotonin re-uptake inhibitors (SSRIs) such as citalopram and escitalopram ((es)citalopram) are among the first-line pharmacological options for depression in Europe and the US, and the use of SSRIs has increased considerably over the years [1,2]. However, reports showed that less than 50% of (es)citalopram users achieved disease symptom remission during their first treatment episode, and prognosis appeared unpredictable [3,4]. Such variable effectiveness may be explained by a large inter-individual pharmacokinetic variability among patients treated with (es)citalopram [5,6]. This variability is known to be caused partly by differences in metabolic activity of drug metabolizing Cytochrome P450 (CYP) enzymes [7].

(Es)citalopram is primarily metabolized by the combination of CYP2C19 and CYP3A4 enzymes, and to a lesser extent by CYP2D6 enzyme [8,9]. Genetic polymorphisms are known to affect the catalytic activity of these enzymes. Some studies have investigated the role of *CYP2C19* and *CYP2D6* polymorphisms on the exposure as well as the clinical impact of (es)citalopram [7,10]. Such interaction between the drug treatment and genetic variation is referred to as drug-gene interaction (DGI) [11]. To the best of our knowledge, no previous studies have explored the impact of the DGI related to CYP3A4 polymorphisms, or its combination with *CYP2C19* polymorphisms, in (es)citalopram treatment. In addition, the concomitant administration of CYP2C19, CYP3A4, and/or CYP2D6 (CYP2C19/3A4/2D6) modulator drugs (inhibitor/inducer) produces a drug-drug-interaction (DDI) with (es)citalopram by affecting blood concentrations and hence modifying its effectiveness [12].

To make it even more complicated for treating physicians, (es)citalopram treatment may be affected by both genetics and drugs that modulate the activity of the metabolic pathways at the same time which potentially affect blood concentration even more unpredictably than DGI and DDI alone [13]. In other words, a drug-drug-gene-interaction (DDGI) is encountered when a DGI coincides with a DDI [14,15]. Generally, DDGIs show more pharmacokinetic diversity than DDIs and DGIs alone, since DDGIs concern several modes of interactions [15,16]. For example, a DDGI may involve the co-existence of a genetic polymorphism and a CYP-inhibitor for one CYP-enzyme or the co-presence of a genetic polymorphism in one or two metabolic pathways and a CYP modulator in another pathway [14,15].

Due to restricted study populations in trials and scarcity of health care databases with a possibility to link genetic and drug dispensing data, large-scale real-world pharmacogenetic studies are lacking on the impact of pharmacogenetic and drug interactions in general. Consequently, recent guidelines have only provided specific recommendations on the management of (es)citalopram-related DGIs and DDIs separately, but a knowledge gap remains regarding the pharmacotherapeutic management of DDGIs [17,18]. The PharmLines Initiative enables the unique linkage of genetic and drug data to perform an inception cohort study in a large population cohort which we used to explore the impact of DDIs, DGIs (specifically *CYP2C19/3A4* polymorphisms), and DDGIs on short-term first-time (es)citalopram therapy [19]. To mirror treatment success, proxy outcomes such as drug switching, dose adjustment, and an early discontinuation after the first prescription of (es)citalopram are used [20,21].

## 2. Methods

### 2.1. Study Design, Setting and Data Sources

This retrospective cohort study was performed using data from the PharmLines Initiative which links the Lifelines cohort and the University of Groningen prescription IADB.nl database, two large databases in the Northern part of the Netherlands [19].

The Lifelines cohort is a three-generation prospective cohort covering 167,729 Dutch participants from the Northern provinces of the Netherlands [22,23]. It was established with the aim to study ‘complex interactions between environmental, phenotypic and genomic factors in the development of chronic diseases and healthy ageing’ [22,23]. The participants from the Lifelines cohort generally represent the characteristics of the adult population of the Northern part of the Netherlands [24]. More comprehensive information about the Lifelines cohort can be found in the publications of Stolk et al. and Scholtens et al. [22,23].

The University of Groningen prescription database IADB.nl collected over 1.2 million prescriptions from 72 pharmacies. The information about gender, date of birth and four-digit postal codes (optional) from 730,000 recorded anonymous patients are available [25]. The prescription information of each participant is recorded such as dispensing date, Anatomical Therapeutic Chemical code (ATC code), quantity, duration, and DDD (defined daily dose) [25]. The participants recorded in the IADB.nl are found to be representative of the general population in the Netherlands as whole [25]. The IADB.nl is a reliable database and has been used in many pharmacoepidemiological studies [26,27,28]

The linking process of these two databases was facilitated by a trusted third party, the Statistic Netherlands. The linkage was performed at the individual level and relied on combined information of postal code, date of birth, and gender. Once the selection process was completed, identifiers from each database were cleared and then, a new unique identifier (pseudoID) was assigned. Using the pseudoID, genetic and prescription information of the participants from the Lifelines cohort and the IADB.nl, respectively, could be combined. Details on the linking process has been published elsewhere [19].

### 2.2. Study Population

Adult Lifelines participants (Caucasian, 18 years and older) with available genetic information (*CYP2C19* and *CYP3A4* genes) and who had their first citalopram (N06AB04) or escitalopram (N06AB10) prescription recorded in the PharmLines Iniative were eligible. Those who were not prescribed any (es)citalopram for at least 180 days before starting their drug dispensing were included. If there were several periods of (es)citalopram dispensing, only the first dispensing period was included in the analysis. Date of the first (es)citalopram prescription was regarded as an index date which indicates the start of follow-up.

### 2.3. Genotyping

Genotyping for single-nucleotide polymorphism (SNP) of *CYP2C19* and *CYP3A4* genes in the Lifelines cohort was performed using the Illumina CytoSNP-12v2 array [22]. The genotype data was imputed by using the Genome of the Netherlands reference panel [22]. The quality of genotyping data was checked using the following requirements i.e., (i) the *p*-value of Hardy-Weinberg equilibrium distribution was > 1 × 10^−4^, (ii) call rate of 95%, and (iii) minor allele frequency (MAF) was > 0.001 [22]. Additionally, principal component analysis was used to detect statistical outliers [22]. More detailed information on the genotyping process can be found in the publication of Scholtens et al. (2014) [22].

*CYP2C19* and *CYP3A4* genotypes were translated to haplotypes, which were used to predict corresponding phenotypes (Table 1, Table 2, Table 3 and Table 4). Relevant haplotypes were selected and genotypes were translated to predicted phenotypes based on available information from the Dutch Pharmacogenetics Working Group (DPWG). Corresponding predicted phenotypes include poor metabolizer (PM), intermediate metabolizer (IM), and normal metabolizer (NM) for CYP2C19 and CYP3A4, and ultra-rapid metabolizer (UM) for CYP2C19.

### 2.4. Definition of Exposures

The exposure groups were defined as (es)citalopram users with a DGI, DDI, or DDGI. Participants who were predicted to be CYP2C19 UM, IM, or PM and/or CYP3A4 IM or PM and were prescribed (es)citalopram without co-prescription of CYP2C19/3A4/2D6 modulators (inhibitors/inducers) were classified as experiencing a DGI. For statistical power reasons, IM and PM groups were pooled into a combined IM/PM group, but we provided a sensitivity analysis for the separated IM and PM groups (Appendix A).

Participants were classified to have a DDI when they were predicted as normal metabolizers (NM) of CYP2C19 and CYP3A4, and at the same time were co-prescribed a CYP2C19 and/or CYP3A4 and/or CYP2D6 modulator during the (es)citalopram treatment within a follow-up time frame of 90 days. A list of clinically relevant CYP2C19/3A4/2D6 modulators was based on *Commentaren Medicatiebewaking* (Health Base, NL) and the Flockhart table^TM^ (Appendix A) [32,33]. Only non-SSRI drugs were included as CYP2C19/3A4/2D6 modulators since our study population consists of first-time (es)citalopram users and it is uncommon to combine this with another SSRI drug in the early phase of drug treatments [34].

DDGI was defined as the occurrence of a DGI and DDI at the same time in which (es)citalopram patients with a CYP2C19/3A4 predicted deviating phenotype received a CYP2C19/3A4/2D6 modulator. The non-exposed reference group was defined as (es)citalopram users with a predicted normal CYP2C19/3A4 and who were not prescribed any CYP2C19/3A4/2D6 modulator during first-time (es)citalopram treatment.

### 2.5. Study Outcomes

Study outcomes were drug switching, dose adjustment, and early discontinuation. The incidence of these outcomes within the time frame of a 90 day follow-up after the index date were identified. This time frame was used since the acute phase treatment of SSRIs is considered to be between 6 and 12 weeks after the start of drug treatment. A previous report indicated that about 70% of antidepressant users stopped their therapy within 90 days [35]. However, since interactions commonly have an immediate effect, the presence of the outcomes within the time frame of a 45 day follow-up after the index date were also explored (Appendix A) [21]. Drug switching was defined as patients having an early discontinuation of (es)citalopram as well as the prescription of another antidepressant, regardless of the class, within 120 days after the index date. The follow-up time frame was expanded for dispensing of other antidepressants from 90 to 120 days after the index date in order to accommodate the possible time gap between the dispensing of (es)citalopram and the new antidepressant [36,37]. Meanwhile, dose adjustment was defined as having a dose reduction or a dose elevation for at least 25% of the first dose within 90 days after the index date. Early discontinuation was defined as discontinuing the prescription of (es)citalopram within 90 days after the index date, having no further re-prescription of (es)citalopram for at least 180 days after the stop date as well as no switching as described previously. In the preliminary analysis the effects of exposure on drug switching and dose reduction were in the same direction, therefore the outcomes were combined. Analysis on the separated outcomes are provided in the Appendix A.

### 2.6. Co-Variates

The following co-variates were recorded to compare groups: age, gender, dose of (es)citalopram at the index date, number of co-prescriptions, and pre-defined drugs as a proxy for certain co-existing comorbidities (Appendix A). (Es)citalopram users had to have at least two prescriptions of these proxy medications within six months before or after the index date to be assumed as having a chronic condition of the potential comorbidities [38]. The presence of NSAIDs co-prescription during (es)citalopram prescription was checked within the time frame of 90 days since the combination of NSAIDs and SSRIs was reported to increase the risk of gastrointestinal bleeding [39]. The potential comorbidities were clustered into one group, namely ‘potential comorbidities,’ in order to increase the power of the calculation. The distribution for each potential comorbidity was compared separately between outcomes and none of them were statistically significant different (*p* < 0.05). Lastly, the distribution of the number of CYP2C19/3A4/2D6 modulator prescriptions during the use of (es)citalopram was compared, since a previous study indicated that the higher the number of CYP2C19/3A4/2D6 modulator prescriptions, the more alteration in the clearance of (es)citalopram [12].

### 2.7. Statistical Analysis

The Chi-square (or Fisher′s exact test) and Mann-Whitney test were used to compare distribution of categorical and skewed distributed continuous variables between outcomes, respectively. Co-variates which differed significantly (*p* < 0.05) were entered into final multivariate logistic regression model to obtain adjusted odds ratio as measure of association (OR). We also provided adjusted *p*-values for false discovery rates due to multiple comparisons using the Benjamini-Hochberg method (*q*-values, with a *q* < 0.05 as the significance threshold). Since some participants did not have dosing information, a complete case analysis in cases of dosing comparison as well as dose adjustment analysis were performed. The baseline characteristics were compared between participants with complete information and participants without dosing information (Appendix A).

## 3. Results

Overall, 316 (es)citalopram users (median 45 years, 63% women) with *CYP2C19* and *CYP3A4* genetic information were available (Figure 1). Baseline characteristics of patients are displayed in Table 5. There were 32.6%, 7.3% and 4.4% of participants to have predicted CYP2C19 IM, PM, and UM, respectively, and there were 17.7% and 1.9% of our sample to have predicted CYP3A4 IM and PM, respectively. After combining both genetic information (regardless the presence of another exposure such as CYP modulators), we found that about 56% of the patients had at least one predicted deviating phenotype of CYP2C19 or CYP3A4. There were about 33%, 6%, 11%, and 4% of the participants having predicted CYP2C19 IM/PM + CYP3A4 NM, CYP2C19 IM/PM + CYP3A4 IM/PM, CYP2C19 NM + CYP3A4 IM/PM, and CYP2C19 UM + CYP3A4 NM/IM, respectively.

Regardless of the number of prescribed CYP modulators, about 18% of the participants were exposed to CYP-modulators during (es)citalopram prescription and most of them were CYP2C19 inhibitors (13.9%). No combination of (es)citalopram with CYP2C19/3A4 inducer alone was identified. Two patients exposed to a combination of CYP modulators (one patient with a CYP2C19 and a CYP2D6 inhibitor, and one patient with a CYP2C19 inhibitor and a CYP3A4 inducer) were excluded since the number was too small to analyze. More than 60% of the participants had at least 20 mg citalopram or 10 mg escitalopram daily (≥ 1 Defined Daily Dose/DDD) at the start of their prescriptions. About 68% of the population had 1 to 2 potential comorbidities and about 78% of them used one to three different type of drugs during (es)citalopram prescription.

The more concomitant the CYP modulator used during (es)citalopram prescription, the more alteration in the (es)citalopram produced [12]. In our sample, we only found less than 10% of them using at least two concomitant CYP modulator at the same pathway. After looking on the combination of exposures (*CYP2C19/3A4* genotypes and CYP modulators) among our study population, we found that 9%, 47%, and 8.5% of participants were exposed to DDIs, DGIs, and DDGIs, respectively. Frequency of each type of DDGIs is presented in Table 6.

There were 25 (7.9%), 7 (2.2%), 80 (25%), and 47 (15%) of (es)citalopram users experiencing drug switching, dose reduction, dose elevation, and early discontinuation, respectively. Number of co-prescriptions seemed to influence the rate of switching (*p* = 0.02). Female gender and a higher dose at the index date are less prevalent in the subgroup that experienced dose elevation of (es)citalopram (*p* = 0.003 and 0.002, respectively) (Table 7).

In our dataset, participants with a predicted CYP2C19 IM phenotype had an increased risk of drug switching and/or dose reduction (aOR: 3.16, 95% CI: 1.41–7.09) but CYP2C19 PM did not show a comparable result (aOR: 0.54, 95% CI: 0.07–4.52) (Table 8). Meanwhile, both CYP2C19 IM and PM had a comparable trend on the risk of early discontinuation (aOR: 0.35, 95% CI: 0.15–0.79 and aOR: 0.41, 95% CI: 0.09–1.89, respectively) (Table 9). 

Furthermore, there was an indication showing that co-presence of CYP3A4 IM/PM in individuals with CYP2C19 IM/PM increased the risk of switching and/or dose reduction of (es)citalopram to a larger extent than the combination of CYP2C19 IM/PM and CYP3A4 NM (aOR: 4.38, 95% CI: 1.22–15.69 and aOR: 2.75, 95% CI: 1.03–7.29, respectively). This effect might be facilitated by the combination of CYP2C19 IM and CYP3A4 IM since there was only one participant with CYP2C19 PM and no participants with CYP3A4 PM experiencing switching or dose reduction (Table 8). Meanwhile, CYP3A4 IM/PM in the co-presence of CYP2C19 NM did not seem to influence the risk of switching and/or dose reduction (aOR: 1.02, 95% CI: 0.19–5.24). No participants with the CYP2C19 UM and CYP3A4 NM/IM combination experienced drug switching and/or dose reduction and no significant association with early discontinuation as well as with dose elevation was observed (Table 8, Table 9 and Table 10).

DDIs seemed to increase the risk of drug switching and/or dose reduction (aOR: 2.82, 95% CI: 0.49–15.97), which was mainly facilitated by the co-presence of CYP2C19 inhibitors, but seemingly not to increase the risk of dose elevation and early discontinuation (Table 8, Table 9 and Table 10).

DDGIs also seemed to increase the risk of drug switching and/or dose reduction (aOR: 2.33, 95% CI: 0.42–12.78). However, there were only two participants with DDGIs experiencing drug switching or dose reduction, consisting of one participant with a DDGI affecting one pathway and the other one with a DDGI affecting two pathways (Appendix A). Consequently, a separated analysis of DDGIs based on the number of pathways affected produced comparable effect sizes (DDGI affecting one pathway: aOR: 2.52, 95% CI: 0.26–24.61; DDGI affecting two pathways: aOR: 2.17, 95% CI: 0.23–20.67).

Overall, there were no associations between the exposures and any outcomes tested reaching the statistical significance threshold of a false discovery rate-adjusted *p*-value (*q* > 0.05).

Analysis using a time frame of 45 days after the index date produced comparable results. CYP2C19 IM increased the risk of switching and the effect size was also larger in combination with CYP3A4 IM/PM (aOR: 6.41, 95% CI: 1.19–34.40) than with CYP3A4 NM (aOR: 2.66, 95% CI: 0.65–10.96). CYP2C19 IM seemingly increased the risk of dose reduction (aOR: 2.69, 95% CI: 0.43–16.96). Lastly, DDIs and DDGIs have a tendency to increase the risk of dose reduction and switching, respectively. Detailed data can be found in Appendix A.

## 4. Discussion

In this explorative inception cohort study, we presented for both CYP2C19 and CYP3A4 the associations of DGI, DDI, and DDGI and the risk of switching or dose adjustments and early discontinuation in the first treatment episode of (es)citalopram. In our relatively small samples, we found an indication that participants with DGI involving predicted CYP2C19 IM tended to experience switching and/or dose reduction, instead of early discontinuation, regardless of the CYP3A4 predicted phenotype. For participants with DGI involving predicted CYP3A4 IM/PM, no influence on switching and/or dose reduction was found. Yet, the effect of CYP2C19 IM might be enhanced by the presence of CYP3A4 IM. DDI and DDGI might be associated with an increased risk of switching or dose reduction, but the associations were not significant with wide confidence intervals.

We found that participants with CYP2C19 IM were more likely to experience switching than those with NM. This is consistent with the study reported by Mrazek et al. which showed that individuals with CYP2C19 reduced catalytic function were less tolerant to citalopram than those with increased catalytic function [40]. We also found that (es)citalopram users with CYP2C19 IM tended to experience dose reductions more than those with CYP2C19 NM. Decreasing the maximum daily dose of (es)citalopram in patients with CYP2C19 IM by 25% of the normal maximum dose is recommended by the DPWG [41]. As a note, we possibly managed to find some associations on CYP2C19 IM and the outcomes because we had a large enough number of (es)citalopram users with the genotype (about 33% of the cohort).

Unfortunately, we did not find any significant association between patients with CYP2C19 PM and UM to the outcomes which was probably due to a limited sample size. Some clinical studies reported that patients with CYP2C19 PM were exposed to (es)citalopram blood concentration to a greater extent than CYP2C19 IM and that patients with CYP2C19 UM had a lower exposure to (es)citalopram compared to CY2C19 NMs [7]. Jukic et al. using about 2000 genotyped persons from the Oslo population found that escitalopram users with CYP2C19 UM and PM (33% of the study population) had a three times higher odds of switching to another antidepressant than those with CYP2C19 NM [20].

To the best of our knowledge, this is the first study to examine the impact of CYP3A4 alone and in combination with CYP2C19 on (es)citalopram treatment. Decreased function of CYP3A4 in the CYP2C19 NM participants did not seem to influence the outcomes, but might have increased the effect of CYP2C19 IM. A comparable trend of effects has been reported for CYP2D6. The effect of the CYP2D6 variant in individuals with CYP2C19 NM on the AUC of citalopram was limited. However, when there was a co-presence of CYP2C19 *1/*2 (IM), the influence of CYP2D6 *1/*4 (IM) became stronger [10].

In our dataset, there were about nine percent of (es)citalopram users exposed to potential DDIs. This might be because about 79% of our study population had at least one comorbidity and therefore, they used other drug(s) which might potentially interact with (es)citalopram. In the Lifelines population, the most prevalent potential CYP2C19 mediated DDI was citalopram and omeprazole [42]. Omeprazole was reported to increase s-citalopram plasma concentration by about 50% to 120% [43,44]. Therefore, it has been recommended that patients with omeprazole or esomeprazole should have a dose adjustment of (es)citalopram [45].

Although we did not find any significant associations between DDGI and the outcomes, this study is the first to explore the impact of complex DDGI on the (es)citalopram treatment at the population level. Generally, DDGI may come in two main scenarios [14,15]. Firstly, it may only affect one metabolic pathway of a drug, for example overlapping conditions between a CYP2C19 UM/IM/PM and a CYP2C19 inhibitor in (es)citalopram users. In this scenario, we might expect that the level of blood concentration of (es)citalopram in an individual with a CYP2C19 UM and a CYP2C19 inhibitor might be different from an individual with a CYP2C19 IM and a CYP2C19 inhibitor [15]. This is because the more the number of active allelic variants in the CYP450, the more difficult for their phenotypes to be converted by the co-presence of inhibitors [46]. The second main scenario is the alteration of two or even three metabolic pathways of a drug. The alteration can be a result of the presence of deviating genotypes in one/two metabolic pathway(s) and the co-presence of CYP modulator in one/two other pathway(s). In this scenario, each possible combination of co-inhibition produced by genetic variation and CYP modulators might result in variation of (es)citalopram concentration in the blood [15]. Therefore, the effect of DDGI can vary depending on the scenario of interactions, the metabolic contribution of the inhibited pathway(s), and the potency of CYP modulators [15]. In this study, since we had only two patients with DDGI experiencing switching (Appendix A), we could not explore more about the impact of the different scenarios on the outcomes. It is plausible because the number of patients with DDI and DGI were limited, we could expect that the number of patients exposed to DDGI is even less. Hence, further study with a larger dataset is needed to provide solid evidence about the impact of DDGI in clinical practice which can be used to support the lack of pharmacotherapeutic management of DDGI in the current guidelines.

Since genotyping is still not a part of routine clinical testing, prescribers often have no indication about the genotype of the patients at the time of prescription. Consequently, the presence of DGI and DDGI related to (es)citalopram, exposing 56% of our study population, is potentially missed by health practitioners. Therefore, in order to avoid DGI and DDGI complex interaction, pre-emptive genotyping, inclusion of genetic information in electronic health records as well as a sophisticated computerized drug interaction surveillance system are needed in clinical practice.

Several potential limitations need to be discussed. First, we did not have data on the blood concentration of (es)citalopram as the best indicator to show the effect of interactions. Consequently, we could not ascertain the effect of DDI/DGI/DDGI on the citalopram metabolism and validate the associations between the exposures and the outcomes. In addition, we did not have information about the genotype status of *CYP2D6*. Therefore, we could not assess the combined effects of *CYP2C19/3A4/2D6* polymorphisms on (es)citalopram efficacy. *CYP2D6* is the most polymorphic CYP enzyme and the prevalence of people with CYP2D6 IM and PM genotypes in the Caucasian population is 40% and 10%, respectively [47]. Therefore, there might be some persons with *CYP2D6* polymorphisms among our participants. Despite its minor metabolic contribution on (es)citalopram disposition, *CYP2D6* polymorphism might corroborate the alteration of citalopram clearance in the presence of *CYP2C19* polymorphism [10]. It was reported in a small DGI study among healthy persons that one participant with CYP2C19 PM and CYP2D6 PM taking citalopram developed severe side effects and was withdrawn early before the study was completed [48]. Therefore, there was a possibility that the co-presence of combined *CYP2C19/3A4/2D6* polymorphisms might produce a substantial effect on citalopram disposition.

Furthermore, though, our study population from the PharmLines database is rather large (6379 participants), the statistical power of the study is relatively low to detect significant associations between multiple exposures as DGI, DDI, DDGI and outcomes. Therefore, the results of this study should be interpreted as hypothesis-generating rather than confirmative to explore potential effects of DGI, DDI, and DDGI on the prognosis of (es)citalopram treatment. Much larger studies are required to further confirm our findings. Lastly, about 30% of our dataset had no information about the dose of (es)citalopram. The missingness may probably not be related to other variables since it may be because pharmacists or pharmacy technicians forgot to include the dose information before sending the prescription data to the IADB.nl. In the baseline comparisons, we found that patients without dosing information were significantly older than those with complete information (Appendix A). Hence, we might underestimate the effect of age on the dose adjustments of (es)citalopram. Among those with complete information, age seemed not to influence the dose elevation or reduction of (es)citalopram (Table 7).

## 5. Conclusions

In conclusion, the predicted CYP2C19 IM phenotype increased the need of drug switching and/or dose reduction, and the co-presence of CYP3A4 IM enhanced these effects. Therefore, when patients receive (es)citalopram, it is important to not only consider the genetic information for CYP2C19 but also the genetic status of CYP3A4 as well.

Despite the fact that DDI and DDGI showed trends towards increased risks of switching and/or dose reduction, no conclusions can be drawn from the results because there were great uncertainties surrounding the estimates. Therefore, further real-world studies with larger samples are needed to confirm the results.

## Figures and Tables

**Figure 1 jpm-10-00256-f001:**
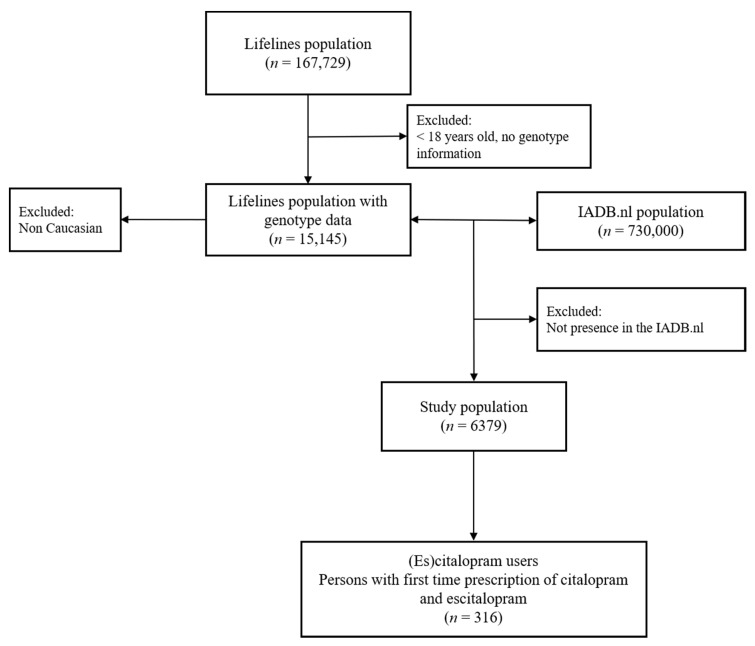
Selection of (es)citalopram first time users.

**Table 1 jpm-10-00256-t001:** Pipeline translation table for *CYP2C19* with haplotypes and their Single Nucleotide Polymorphisms (SNPs) information.

Haplotype Name	Gene	rsID	Reference Sequence	Variant. Start	Variant. Stop	Reference. Allele	Variant. Allele	Type
CYP2C19*1	*CYP2C19*	rs3758581	10	96602622	96602622	G	-	single
CYP2C19*1	*CYP2C19*	rs12769205	10	96535123	96535123	A	-	single
CYP2C19*1	*CYP2C19*	rs28399504	10	96522462	96522462	A	-	single
CYP2C19*1	*CYP2C19*	rs41291556	10	96535172	96535172	T	-	single
CYP2C19*1	*CYP2C19*	rs11188072	10	96519060	96519060	C	-	single
CYP2C19*2	*CYP2C19*	rs12769205	10	96535123	96535123	A	G	single
CYP2C19*4	*CYP2C19*	rs28399504	10	96522462	96522462	A	G	single
CYP2C19*5/7	*CYP2C19*	rs3758581	10	96602622	96602622	G	A	single
CYP2C19*8	*CYP2C19*	rs41291556	10	96535172	96535172	T	C	single
CYP2C19*17	*CYP2C19*	rs11188072	10	96519060	96519060	C	T	single

**Table 2 jpm-10-00256-t002:** Pipeline translation table for *CYP3A4* with haplotypes and their SNP information.

Haplotype. Name	Gene	rsID	Reference Sequence	Variant. Start	Variant. Stop	Reference. Allele	Variant. Allele	Type
CYP3A4*1A	*CYP3A4*	rs2740574	7	99382095	99382095	T	-	single
CYP3A4*1A	*CYP3A4*	rs2242480	7	99361465	99361465	C	-	single
CYP3A4*1A	*CYP3A4*	rs35599367	7	99366315	99366315	G	-	single
CYP3A4*1B	*CYP3A4*	rs2740574	7	99382095	99382095	T	C	single
CYP3A4*1G	*CYP3A4*	rs2242480	7	99361465	99361465	C	T	single
CYP3A4*22	*CYP3A4*	rs35599367	7	99366315	99366315	G	A	single

**Table 3 jpm-10-00256-t003:** The translation of *CYP2C19* and *CYP3A4* haplotypes to their predicted metabolic activity.

Gene	Haplotype	Metabolic Function	Reference
*CYP2C19*	*CYP2C19**1	Normal	[29]
	*CYP2C19**2	No	[29]
	*CYP2C19**4	No	[29]
	*CYP2C19**5/7	No	[29]
	*CYP2C19**8	No	[29]
	*CYP2C19**17	Increased	[29]
*CYP3A4*	*CYP3A4**1A	Normal	[29]
	*CYP3A4**1B	Normal	[30]
	*CYP3A4**1G	Decreased	[31]
	*CYP3A4**22	Decreased	[29]

**Table 4 jpm-10-00256-t004:** The translation of *CYP2C19* and *CYP3A4* haplotype combinations to their predicted phenotypes.

*CYP2C19*	No	Normal	Increased	*CYP3A4*	Decreased	Normal
No	PM	IM	IM	Decreased	PM	IM
Normal	IM	NM	NM	Normal	IM	NM
Increased	IM	NM	UM			

NM: Normal Metabolizer. IM: Intermediate Metabolizer. PM: Poor Metabolizer. UM: Ultrarapid Metabolizer.

**Table 5 jpm-10-00256-t005:** Characteristics of patients starting (es)citalopram (*n* = 316).

Variabels	*N*	%
Gender (*n* women, %)	200	63.3
Age in years, median (IQR)	45	14
CYP2C19 Phenotypes		
CYP2C19 NM (*n*, %)	176	55.7
CYP2C19 IM (*n*, %)	103	32.6
CYP2C19 PM (*n*, %)	23	7.3
CYP2C19 UM (*n*, %)	14	4.4
CYP3A4 Phenotypes		
CYP3A4 NM (*n*, %)	254	80.4
CYP3A4 IM (*n*, %)	56	17.7
CYP3A4 PM (*n*, %)	6	1.9
Combination of CYP2C19 & CYP3A4 Phenotypes		
CYP2C19 NM + CYP3A4 NM (*n*, %)	140	44.3
CYP2C19 IM/PM + CYP3A4 NM (*n*, %)	104	32.9
CYP2C19 IM/PM + CYP3A4 IM/PM (*n*, %)	20	6.3
CYP2C19 NM + CYP3A4 IM/PM (*n*, %)	36	11.4
CYP2C19 UM + CYP3A4 NM/IM (*n*, %)	14	4.4
Type of CYP modulator combination		
No inhibitor or inducer of CYP2C19/3A4/2D6	260	82.3
CYP2C19 inhibitor alone (*n*, %)	44	13.9
CYP3A4 inhibitor alone (*n*, %)	4	1.3
CYP2D6 inhibitor alone (*n*, %)	6	1.9
CYP2C19 inhibitor + CYP2D6 inhibitor (*n*, %) *	1	0.3
CYP2C19 inhibitor + CYP3A4 inducer (*n*, %) *	1	0.3
DDD at start of citalopram and escitalopram		
DDD < 1 (*n*, %)	25	7.9
DDD >= 1 (*n*, %)	197	62.3
No dose information (*n*, %)	94	29.7
Potential comorbidities		
No comorbidity (*n*, %)	65	20.6
1–2 potential comorbidities (*n*, %)	216	68.3
≥3 potential comorbidities (*n*, %)	35	11.1
Number of co-prescriptions during (es)citalopram		
1–3 type of drugs (*n*, %)	247	78.2
>3 type of drugs (*n*, %)	69	21.8
Number of CYP modulator during (es)citalopram		
No CYP modulator (*n*, %)	260	82.3
1 CYP modulator (*n*, %)	27	8.5
≥2 CYP modulator (*n*, %)	29	9.2
Combined exposures		
*No exposures*		
CYP2C19 NM + CYP3A4 NM + No CYP Modulator (*n*, %)	111	35.1
*DDI*		
CYP2C19 NM + CYP3A4 NM + Yes CYP Modulator (*n*, %)	29	9.2
*DGI*		
CYP2C19 IM/PM + CYP3A4 NM + No CYP Modulator (*n*, %)	89	28.2
CYP2C19 IM/PM + CYP3A4 IM/PM + No CYP Modulator (*n*, %)	20	6.3
CYP2C19 NM + CYP3A4 IM/PM + No CYP Modulator (*n*, %)	29	9.2
CYP2C19 UM + CYP3A4 NM/IM + No CYP Modulator (*n*, %)	11	3.5
*DDGI* (*n*, %)	27	8.5

* Excluded. NM: Normal Metabolizer. IM: Intermediate Metabolizer. PM: Poor Metabolizer. UM: Ultrarapid Metabolizer. DDD: Defined Daily Dose. CYP: Cytochrome P450.

**Table 6 jpm-10-00256-t006:** Frequency of DDGI (overlapping condition of DDI and DGI).

CYP2C19 Phenotype	CYP3A4 Phenotype	CYP2C19 Inhibitor	CYP3A4 Inhibitor	CYP2D6 Inhibitor	CYP2C19 Inducer	CYP3A4 Inducer	*N*	%
One pathway								
UM/IM/PM	NM	Y	N	N	N	N	14	51.8
Two pathways								
IM	IM	Y	N	N	N	N	2	7.4
IM	NM	N	Y	N	N	N	2	7.4
IM	NM	N	N	Y	N	N	2	7.4
NM	IM/PM	Y	N	N	N	N	6	22.2
NM	IM	N	N	Y	N	N	1	3.7
SUM							27	

NM: Normal Metabolizer. IM: Intermediate Metabolizer. PM: Poor Metabolizer. UM: Ultrarapid Metabolizer. Y: Yes. N: No.

**Table 7 jpm-10-00256-t007:** Baseline comparisons.

Variables	Switching *		*p*-Value	Decreased Dose ^#^		*p*-Value	Increased Dose ^#^		*p*-Value	Discontinuation *		*p*-Value
	Yes (*n =* 25)	No (*n =* 279)		Yes (*n =* 7)	No (*n =* 213)		Yes (*n =* 80)	No (*n =* 140)		Yes (*n =* 47)	No (*n =* 257)	
Gender (*n* Women)	15	177	0.73	5	133	1.00	40	98	0.003	29	163	0.82
Age in years (median, IQR)	41	45	0.68	39	42	0.38	43.5	41	0.92	48	44	0.03
DDD at start (*n* DDD ≥1)	18	177	1.00	7	188	1.00	64	131	0.002	31	164	0.57
Potential comorbidities (*n* Yes)												
No comorbidity (*n*)	3	60	0.18	3	36	0.22	16	23	0.79	9	54	0.71
1–2 potential comorbidities (*n*)	21	185		4	152		55	101		34	172	
≥3 potential comorbidities (*n*)	1	34		0	25		9	16		4	31	
N of co-prescriptions												
1–3 (*n*)	24	213	0.02	7	166	0.35	63	110	0.97	38	199	0.60
>3 (*n*)	1	66		0	47		17	30		9	58	
N of CYP modulator prescriptions												
No (*n*)	22	229	0.92	6	177	0.73	69	114	0.62	41	210	0.64
1 (*n*)	2	25		1	18		5	14		4	23	
≥2 (*n*)	1	25		0	18		6	12		2	24	

* No start/stop date = 10; ^#^ no dose information = 94. NM: Normal Metabolizer. IM: Intermediate Metabolizer. PM: Poor Metabolizer. UM: Ultrarapid Metabolizer. DDD: Defined Daily Dose. CYP: Cytochrome P450. N: Number. IQR: Interquartile Range.

**Table 8 jpm-10-00256-t008:** Association between DDI, DGI, and DDGI with drug switching and/or dose reduction.

Variables	Switching and/or Dose Reduction		Univariate Analysis			Multivariate Analysis		
	Yes (*n =* 31, %)	No (*n =* 273, %)	OR (95%CI)	*p*-Value	*q*-Value	aOR (95%CI)	*p*-Value	*q*-Value
CYP2C19 & CYP3A4 predicted phenotypes								
CYP2C19 predicted phenotypes *								
CYP2C19 NM	12 (38.7)	157 (57.5)	Ref.			Ref.		
CYP2C19 IM	18 (58.1)	82 (30)	2.87 (1.32–6.25)	0.01	0.08	3.16 (1.41–7.09)	0.005	0.06
CYP2C19 PM	1 (3.2)	20 (7.3)	0.65 (0.08–5.30)	0.69	0.90	0.54 (0.07–4.52)	0.57	0.68
CYP2C19 UM	0 (0)	14 (5.1)	*NA*			*NA*		
CYP3A4 predicted phenotypes **								
CYP3A4 NM	23 (74.2)	220 (80.6)	Ref.					
CYP3A4 IM	8(25.8)	47 (17.2)	1.63 (0.69–3.86)	0.27	0.54	1.37 (0.55–3.39)	0.50	0.67
CYP3A4 PM	0 (0)	6 (2.2)	*NA*			*NA*		
Combination of predicted phenotypes ***								
CYP2C19 NM + CYP3A4 NM	9 (29)	125 (45.8)	Ref.			Ref.		
CYP2C19 IM/PM + CYP3A4 NM	14 (45.2)	85 (31.1)	2.29 (0.95–5.52)	0.07	0.17	2.35 (0.96–5.76)	0.06	0.14
CYP2C19 IM/PM + CYP3A4 IM/PM	5 (16.1)	17 (6.2)	4.08 (1.22–13.63)	0.02	0.08	3.46 (1.02–11.75)	0.05	0.14
CYP2C19 NM + CYP3A4 IM/PM	3 (9.7)	32 (11.7)	1.30 (0.33–5.09)	0.70	0.90	1.11 (0.28–4.43)	0.88	0.96
CYP2C19 UM + CYP3A4 NM/IM	0 (0)	14 (5.1)	*NA*			*NA*		
CYP modulator ^#^								
No inhibitor/inducer of CYP2C19/3A4/2D6	27 (87.1)	224 (82.1)	Ref.			Ref.		
CYP2C19 inhibitor	4 (12.9)	40 (14.7)	0.83 (0.27–2.49)	0.74	0.90	2.36 (0.67–8.32)	0.18	0.36
CYP3A4 inhibitor	0 (0)	4 (1.5)	*NA*			*NA*		
CYP2D6 inhibitor	0 (0)	5 (1.8)	*NA*			*NA*		
Combined exposures ^								
*No exposures*								
CYP2C19 NM + CYP3A4 NM + No CYP Modulator	7 (22.6)	101 (37)	Ref.			Ref.		
*DDI*								
CYP2C19 NM + CYP3A4 NM + Yes CYP Modulator	2 (6.5)	24 (8.8)	1.20 (0.24–6.16)	0.83	0.90	2.82 (0.49–15.97)	0.24	0.41
*DGI*								
CYP2C19 IM/PM + CYP3A4 NM + No CYP Modulator	13 (41.9)	71 (26)	2.64 (1.00–6.95)	0.05	0.15	2.75 (1.03–7.29)	0.04	0.14
CYP2C19 IM/PM + CYP3A4 IM/PM + No CYP Modulator	5 (16.1)	15 (5.5)	4.81 (1.35–17.12)	0.02	0.08	4.38 (1.22–15.69)	0.02	0.12
CYP2C19 NM + CYP3A4 IM/PM + No CYP Modulator	2 (6.5)	26 (9.5)	1.11 (0.22–5.66)	0.90	0.90	1.02 (0.19–5.24)	0.98	0.98
CYP2C19 UM + CYP3A4 NM/IM + No CYP Modulator	0 (0)	11 (4)	*NA*			*NA*		
*DDGI*	2 (6.5)	25 (9.2)	1.15 (0.23–5.89)	0.86	0.90	2.33 (0.42–12.78)	0.33	0.49

Adjusted for: * CYP3A4 phenotypes, CYP modulator & N of co-prescriptions; ** CYP2C19 phenotypes, CYP modulator & N of co-prescriptions; *** CYP modulator & N of co-prescriptions; ^#^ CYP2C19 & CYP3A4 phenotypes & N of co-prescriptions; ^ N of co-prescriptions. NM: Normal Metabolizer. IM: Intermediate Metabolizer. PM: Poor Metabolizer. UM: Ultrarapid Metabolizer. DDI: Drug-Drug interaction. DGI: Drug-Gene Interaction. DDGI: Drug-Drug-Gene Interaction (overlapping condition of DDI and DGI, please see Table 6: Frequency of DDGI). CYP: Cytochrome P450. NA: Not Available. OR: Odds Ratio. aOR: Adjusted Odds ratio. CI: Confidence Interval. Ref.: Reference.

**Table 9 jpm-10-00256-t009:** Association between DDI, DGI, and DDGI with early discontinuation.

Variables	Early Discontinuation		Univariate Analysis			Multivariate Analysis		
	Yes (*n =* 47, %)	No (*n =* 257, %)	OR (95%CI)	*p*-Value	*q*-Value	aOR (95%CI)	*p*-Value	*q*-Value
CYP2C19 & CYP3A4 predicted phenotypes								
CYP2C19 phenotypes *								
CYP2C19 NM	33 (70.2)	136 (52.9)	Ref.			Ref.		
CYP2C19 IM	9 (19.1)	91 (35.4)	0.41 (0.19–0.89)	0.03	0.45	0.35 (0.15–0.79)	0.01	0.15
CYP2C19 PM	2 (4.3)	19 (7.4)	0.43 (0.09–1.96)	0.28	0.50	0.41 (0.09–1.89)	0.25	0.54
CYP2C19 UM	3 (6.4)	11 (4.3)	1.12 (0.29–4.26)	0.86	0.86	1.24 (0.32–4.88)	0.75	0.75
CYP3A4 phenotypes **								
CYP3A4 NM	36 (76.6)	207 (80.5)	Ref.			Ref.		
CYP3A4 IM	11 (23.4)	44 (17.1)	1.44 (0.68–3.04)	0.34	0.51	1.29 (0.59–2.84)	0.51	0.59
CYP3A4 PM	0 (0)	6 (2.3)	*NA*			*NA*		
Combination of predicted phenotypes ***								
CYP2C19 NM + CYP3A4 NM	24 (51.1)	110 (42.8)	Ref.			Ref.		
CYP2C19 IM/PM + CYP3A4 NM	10 (21.3)	89 (34.6)	0.52 (0.23–1.13)	0.09	0.45	0.45 (0.20–1.02)	0.06	0.35
CYP2C19 IM/PM + CYP3A4 IM/PM	1 (2.1)	21 (8.2)	0.22 (0.03–1.70)	0.15	0.45	0.17 (0.02–1.39)	0.10	0.36
CYP2C19 NM + CYP3A4 IM/PM	9 (19.1)	26 (10.1)	1.59 (0.66–3.81)	0.30	0.50	1.43 (0.58–3.53)	0.44	0.59
CYP2C19 UM + CYP3A4 NM/IM	3 (6.4)	11 (4.3)	1.25 (0.32–4.83)	0.75	0.80	1.43 (0.36–5.69)	0.61	0.65
CYP modulator ^#^								
No inhibitor/inducer of CYP2C19/3A4/2D6	41 (87.2)	210 (81.7)	Ref.			Ref.		
CYP2C19 inhibitor alone	6 (12.8)	38 (14.8)	0.81 (0.32–2.04)	0.65	0.80	0.68 (0.26–1.75)	0.42	0.59
CYP3A4 inhibitor alone	0 (0)	4 (1.6)	*NA*			*NA*		
CYP2D6 inhibitor alone	0 (0)	5 (1.9)	*NA*			*NA*		
Combined exposures ^								
*No exposures*								
CYP2C19 NM + CYP3A4 NM + No CYP Modulator	20 (42.6)	88 (34.2)	Ref.			Ref.		
*DDI*								
CYP2C19 NM + CYP3A4 NM + Yes CYP Modulator	4 (8.5)	22 (8.6)	0.80 (0.25–2.58)	0.71	0.80	0.67 (0.20–2.21)	0.51	0.59
*DGI*								
CYP2C19 IM/PM + CYP3A4 NM + No CYP Modulator	9 (19.1)	75 (29.2)	0.53 (0.23–1.23)	0.14	0.45	0.44 (0.19–1.06)	0.07	0.35
CYP2C19 IM/PM + CYP3A4 IM/PM + No CYP Modulator	1 (2.1)	19 (7.4)	0.23 (0.03–1.83)	0.17	0.45	0.19 (0.02–1.53)	0.12	0.36
CYP2C19 NM + CYP3A4 IM/PM + No CYP Modulator	8 (17)	20 (7.8)	1.76 (0.68–4.56)	0.25	0.50	1.52 (0.57–4.04)	0.41	0.59
CYP2C19 UM + CYP3A4 NM/IM + No CYP Modulator	3 (6.4)	8 (3.1)	1.65 (0.40–6.78)	0.49	0.67	1.89 (0.45–8.07)	0.39	0.59
*DDGI*	2 (4.3)	25 (9.7)	0.35 (0.08–1.61)	0.18	0.45	0.38 (0.08–1.75)	0.21	0.53

Adjusted for: * CYP3A4 phenotypes, CYP modulator & age; ** CYP2C19 phenotypes, CYP modulator & age; *** CYP modulator & age; ^#^ CYP2C19 & CYP3A4 phenotypes & age; ^ age. NM: Normal Metabolizer. IM: Intermediate Metabolizer. PM: Poor Metabolizer. UM: Ultrarapid Metabolizer. DDI: Drug-Drug interaction. DGI: Drug-Gene Interaction. DDGI: Drug-Drug-Gene Interaction (overlapping condition of DDI and DGI, please see Table 6: Frequency of DDGI). CYP: Cytochrome P450. NA: Not Available. OR: Odds Ratio. aOR: Adjusted Odds ratio. CI: Confidence Interval. Ref.: Reference.

**Table 10 jpm-10-00256-t010:** Association between DDI, DGI, and DDGI with dose elevation.

Variables	Dose Elevation		Univariate Analysis			Multivariate Analysis		
	Yes (*n =* 80, %)	No (*n =* 140, %)	OR (95%CI)	*p*-Value	*q*-Value	aOR (95%CI)	*p*-Value	*q*-Value
CYP2C19 & CYP3A4 predicted phenotypes								
CYP2C19 predicted phenotypes *								
CYP2C19 NM	51 (63.7)	67 (47.9)	Ref.			Ref.		
CYP2C19 IM	23 (28.7)	56 (40)	0.54 (0.29–0.99)	0.05	0.45	0.59 (0.31–1.12)	0.11	0.54
CYP2C19 PM	4 (5)	10 (7.1)	0.53 (0.16–1.77)	0.29	0.61	0.56 (0.16–2.02)	0.38	0.54
CYP2C19 UM	2 (2.5)	7 (5)	0.37 (0.07–1.88)	0.23	0.61	0.35 (0.07–1.85)	0.22	0.54
CYP3A4 predicted phenotypes **								
CYP3A4 NM	61 (76.3)	114 (81.4)	Ref.			Ref.		
CYP3A4 IM	17 (21.3)	24 (17.1)	1.32 (0.66–2.65)	0.43	0.61	1.48 (0.70–3.12)	0.30	0.54
CYP3A4 PM	2 (2.5)	2 (1.4)	1.87 (0.26–13.59)	0.54	0.66	1.27 (0.15–10.64)	0.82	0.87
Combination of predicted phenotypes ***								
CYP2C19 NM + CYP3A4 NM	40 (50)	56 (40)	Ref.			Ref.		
CYP2C19 IM/PM + CYP3A4 NM	21 (26.3)	52 (37.1)	0.56 (0.29–1.08)	0.08	0.45	0.69 (0.35–1.36)	0.28	0.54
CYP2C19 IM/PM + CYP3A4 IM/PM	6 (7.5)	14 (10)	0.60 (0.21–1.69)	0.34	0.61	0.57 (0.19–1.68)	0.31	0.54
CYP2C19 NM + CYP3A4 IM/PM	11 (13.8)	11 (7.9)	1.40 (0.55–3.54)	0.48	0.63	1.66 (0.62–4.49)	0.31	0.54
CYP2C19 UM + CYP3A4 NM/IM	2 (2.5)	7 (5)	0.40 (0.08–2.03)	0.27	0.61	0.41 (0.08–2.18)	0.29	0.54
CYP modulator ^#^								
No inhibitor/inducer of CYP2C19/3A4/2D6	69 (86.3)	114 (81.4)	Ref.			Ref.		
CYP2C19 inhibitor alone	9 (11.3)	21 (15)	0.71 (0.31–1.63)	0.42	0.61	0.80 (0.33–1.95)	0.63	0.76
CYP3A4 inhibitor alone	2 (2.5)	2 (1.4)	1.65 (0.23–11.99)	0.62	0.70	2.75 (0.37–20.74)	0.33	0.54
CYP2D6 inhibitor alone	0 (0)	3 (2.1)	*NA*			*NA*		
Combined exposures ^								
*No exposures*								
CYP2C19 NM + CYP3A4 NM + No CYP Modulator	33 (41.3)	46 (32.9)	Ref.			Ref.		
*DDI*								
CYP2C19 NM + CYP3A4 NM + Yes CYP Modulator	7 (8.8)	10 (7.1)	0.98 (0.34–2.83)	0.96	0.96	1.03 (0.34–3.12)	0.96	0.96
*DGI*								
CYP2C19 IM/PM + CYP3A4 NM + No CYP Modulator	18 (22.5)	42 (30)	0.59 (0.29–1.22)	0.16	0.61	0.69 (0.33–1.45)	0.33	0.54
CYP2C19 IM/PM + CYP3A4 IM/PM + No CYP Modulator	6 (7.5)	13 (9.3)	0.64 (0.22–1.87)	0.42	0.61	0.64 (0.21–1.91)	0.42	0.55
CYP2C19 NM + CYP3A4 IM/PM + No CYP Modulator	10 (12.5)	9 (6.4)	1.55 (0.57–4.23)	0.39	0.61	1.60 (0.56–4.56)	0.37	0.54
CYP2C19 UM + CYP3A4 NM/IM + No CYP Modulator	2 (2.5)	4 (2.9)	0.69 (0.12–4.03)	0.69	0.73	0.72 (0.12–4.35)	0.72	0.82
*DDGI*	4 (5)	16 (11.4)	0.35 (0.11–1.14)	0.08	0.45	0.48 (0.14–1.61)	0.23	0.54

Adjusted for: * CYP3A4 phenotypes, CYP modulator, gender & dose at start; ** CYP2C19 phenotypes, CYP modulator, gender & dose at start; *** CYP modulator, gender & dose at start; ^#^ CYP2C19 & CYP3A4 phenotypes, gender & dose at start; ^ gender & dose at start.NM: Normal Metabolizer. IM: Intermediate Metabolizer. PM: Poor Metabolizer. UM: Ultrarapid Metabolizer. DDI: Drug-Drug interaction. DGI: Drug-Gene Interaction. DDGI: Drug-Drug-Gene Interaction (overlapping condition of DDI and DGI, please see Table 6: Frequency of DDGI) CYP: Cytochrome P450. NA: Not Available. OR: Odds Ratio. aOR: Adjusted Odds ratio. CI: Confidence Interval. Ref.: Reference.

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
