# Peer review of "Impact of Drug-Gene-Interaction, Drug-Drug-Interaction, and Drug-Drug-Gene-Interaction on (es)Citalopram Therapy: The PharmLines Initiative"

_jpm, 2020, doi:10.3390/jpm10040256_

Round 1

Reviewer 1 Report

The manuscript is in line with the currently popular trend in  medicine to look how the genes and other drugs affect treatment regimen. I congratulate to authors for this interesting work.

Data basicly supports the conclusions form existing literature, however there are some issues that I want to point:

  1. Did you check the HWE within the sample?
  2. How did the authors control for type I errors? If such analyses was not conducted please use false discovery rate (FDR) approach
  3. If power analyses were considered, can you provide data on these analyses which made you to join IM and PM groups?
  4. The coding for genotype=phenotype provided in suppl should be within the main text, as without this the tables and text are not easy to follow
  5. Please explain abbrev. under tables
  6. You mentioned PPIs within the discusion but you did not show analyses for persons with this drug intake only

Reviewer 2 Report

The publication is in general well written but requires some improvement. Citalopram is metabolized via CYP2C9, 2A4 and 2D6. As 2D6 is a typical polymorphic enzyme and 2D6 modulators were included in the evaluation more emphasis/explanation should be put on the non-availability of the individual genotype information. DDGI was emphasized as the major gap in the introduction. However, the used dataset is insufficient/too small to do a scientifically based evaluation. Either the introduction or the discussion requires improvement on that. The result section requires more details what data and how the data are presented (see detailed comments in attached document). The discussion touches all parts from the results. Also limitations are presented but not really discussed in details. It is a huge list of limitations (less but with more detailed discussion would be benefitial) which gives the impression that there are doubts on the validity of the study. This should be improved. Finally, the outcome of the main goal should be clearly stated in the conclusion and what is the take-away message for the reader.

Round 2

Reviewer 1 Report

Congratulate for the very nice paper.

The only comment I have is to put Tables 1&2 in the supplements (now too many tables are present within the text).

Reviewer 2 Report

The paper distincly improved with the revision. I have no further comments.